# In Vitro Activity of Quaternary Ammonium in *Prototheca* Isolated from Clinical Bovine Mastitis Identified by Mass Spectrometry and PCR Sequencing of the *cytb* Gene Marker

**DOI:** 10.3390/ani13203286

**Published:** 2023-10-21

**Authors:** Marcelo Fagali Arabe Filho, Tomasz Jagielski, Angelika Proskurnicka, Marcos Veiga dos Santos, Carlos Eduardo Fidelis, Felipe Freitas Guimarães, Simony Trevizan Guerra, Sâmea Fernandes Joaquim, Mateus de Souza Ribeiro Mioni, José Carlos de Figueiredo Pantoja, Helio Langoni, Luísa Fernanda García Sanchez, Marcio Garcia Ribeiro

**Affiliations:** 1Department of Animal Production and Preventive Veterinary Medicine, School of Veterinary Medicine and Animal Science, Sao Paulo State University—UNESP, Botucatu 19806-900, SP, Brazil; fagali.arabe@unesp.br (M.F.A.F.); felipefreitasguimaraes@hotmail.com (F.F.G.); simony.guerra@gmail.com (S.T.G.); sameajoaquim@gmail.com (S.F.J.); jose.pantoja@unesp.br (J.C.d.F.P.); helio.langoni@unesp.br (H.L.); luisa.garcia@unesp.br (L.F.G.S.); 2Department of Medical Microbiology, Institute of Microbiology, Faculty of Biology, University of Warsaw, 00-927 Warsaw, Poland; t.jagielski@uw.edu.pl (T.J.); aproskurnicka@gmail.com (A.P.); 3Department of Animal Nutrition and Production, School of Veterinary Medicine and Animal Science, University of Sao Paulo—USP, Pirassununga 05508-220, SP, Brazil; mveiga@usp.br (M.V.d.S.); carlosfidelis@usp.br (C.E.F.); 4Department of Pathology, Reproduction and One Health, Sao Paulo State University—UNESP, Jaboticabal 14884-900, SP, Brazil; mateus.mioni@unesp.br

**Keywords:** *Prototheca bovis*, algaecide compounds, intramammary protothecosis, molecular typing, clinical severity scoring

## Abstract

**Simple Summary:**

*Prototheca* algae are among the primary agents of bovine mastitis, incurring significant losses to the dairy industry related to a decrease in milk quality and yield, premature culling, and early replacement of infected cows. To date, there is no effective therapy for bovine mammary protothecosis. Here, the in vitro algaecide activity of quaternary ammonium (QA), a sanitizer used for animal and human purposes, was observed in *Prototheca* algae isolated from mastitic cows, indicating the potential use of this compound as an antiseptic/disinfectant for milking facilities and surrounding environmental areas.

**Abstract:**

The in vitro algaecide activity of quaternary ammonium (QA) against *Prototheca* isolated from bovine clinical mastitis was investigated, in which the clinical severity was scored, milk samples were subjected to microbiological culture, and algal species were identified by molecular typing. A total of 4275 milk clinical samples of different cows from ten large dairy farms were used. Forty-four (1%) samples of cows from three dairy farms yielded growth of *Prototheca*, of which 88.6% (39/44) were identified as *Prototheca bovis* and 11.3% (5/44) as *Prototheca* sp. by MALDI-TOF MS, whereas 100% of the isolates were identified as *P. bovis* using PCR sequencing of the *cytb* gene. Among cows for which clinical severity scoring was available, 78.8% (26/33) and 21.2% (7/33) had mild and moderate infections, respectively, whereas no animal showed severe clinical signs. The algaecide activity of QA in *Prototheca* was observed in low concentrations among all isolates, in 20.4% (9/44) at 35 ppm, 36.4% (16/44) at 17 ppm, and 43.2% (19/44) at an 8 ppm, in addition to activity on three reference *Prototheca* strains. Overall, the study highlights the predominance of *P. bovis* as the causative agent of algal mastitis in bovines. *Prototheca* induced abnormalities preponderantly in the milk and mammary gland tissue of cows, and to our knowledge, our study is the first to apply clinical severity scoring in protothecal mastitis. In addition, the study underlines the activity of QA in low concentrations against *Prototheca*, indicating its potential use as an antiseptic/disinfectant in milking facilities and dairy environments.

## 1. Introduction

Microalgae of the genus *Prototheca* are eukaryotic, saprophytic, and ubiquitous organisms that reproduce asexually by endosporulation [1]. These algae lost their photosynthetic ability during evolution, which induced heterotrophic metabolism, allowing the development of opportunistic infections in animals and humans [2].

The current taxonomy of the algae has been built based on the mitochondrial *cytb* gene, subdividing the genus *Prototheca* into 14 well-defined species, namely: *P. ciferrii* (formerly *P. zopfii* genotype 1 or biovar 1), *P. bovis* (formerly *P. zopfii* genotype 2 or biovar 2), *P. blaschkeae* (formerly biovar 3), *P. wickerhamii*, *P. cutis*, *P. miyajii*, *P. moriformis* (formerly *P. ulmea*), *P. stagnora*, *P. tumulicola*, *P. cookei*, *P. pringsheimii*, *P. xanthoriae*, and *P. cerasi* [1]. Recently, another four *Prototheca* species have been described, namely *P. paracutis*, *P. fontanea*, *P. lentecrescens*, and *P. vistulensis* [3,4]. Among all *Prototheca* species, *P. bovis* and, less frequently, *P. blaschkeae* have been recognized as the causative agents of bovine mammary infections worldwide [5].

Bovine mastitis is the most common clinical manifestation of protothecosis among domestic animals [1]. An increasing occurrence of intramammary protothecosis has been reported in several countries [1,6,7,8,9,10,11,12,13], including those having modern dairy herds with well-controlled management of contagious agents [14].

*Prototheca* species inhabit wet sources containing feces, decomposing organic matter, stagnant water, and animal bedding, as well as contaminating milking utensils, which favor mammary infections in cows from the dairy environment and surrounding areas [15]. Mammary protothecosis causes significant losses to dairy industries, through a significant reduction in milk yield and quality, increased milk cell counts, loss of mammary quarters, premature culling of animals, costs of veterinary services, and early replacement of animals [5].

Mammary protothecosis has been associated predominantly with clinical infections and, less frequently, subclinical cases, with a chronic evolution, usually restricted to clinical abnormalities in the milk and mammary gland [16].

In the last decade, the clinical severity of bovine mammary infections has been scored as mild, moderate, and severe cases [17] and investigated mainly among environmental agents [18,19]; although, this scoring method for clinical cases has not been investigated in protothecal mammary infections.

To date, no therapy protocol has been fully effective in controlling *Prototheca*-induced infections in dairy cows; a fact that has prompted a set of in vitro studies focused on the algaecide activity of a wide range of antimicrobials [20,21], antifungals [22], sanitizers [6,15,23,24,25,26], natural extracts [27], essential oils [28], polypeptides [29], nanoparticles [30], photodynamic therapy [31], and herbicides [32].

Quaternary ammonium compounds (QAC) are cationic surfactants commonly used as antiseptics and disinfectants in human and veterinary medicine due to their microbicidal action [33], which denatures proteins of microbial cell membranes and cytoplasm [34]. In addition, they possess other properties, such as low toxicity, good stability, biodegradability, lack of odor, little corrosiveness, and low skin irritation potential toward the skin and mucous membranes [35]. These characteristics have enabled their use in domestic, agricultural, and hospital disinfection applications, as well as in food industries, animal facilities, and swimming pools [36]. Nonetheless, despite their well-known microbicidal action, no in vitro studies have investigated the algaecide effect of quaternary ammonium (QA) used in swimming pools against *Prototheca* species molecularly identified and isolated from clinical mammary infections in cattle.

Given the increase in protothecal mastitis worldwide, the lack of effective therapy for mammary infections, and severe losses to the dairy industry related to bovine mastitis-related *Prototheca* species, we investigated the in vitro algaecide activity of QA against *Prototheca* isolated from clinical bovine mastitis, identified at the species-level based on mass spectrometry and PCR sequencing of the *cytb* gene.

## 2. Results and Discussion

Of the total milk samples cultured, 1% (44/4275) yielded growth of *Prototheca* isolates. The algae originated from only three (3/10 = 30%) farms studied (Figure 1), giving within-farm prevalences of mammary protothecosis of 3.7% (8/212), 4.6% (33/721), and 4.5% (3/66) in farms B, E, and J, respectively (Table 1).

All *Prototheca* isolates were cultured from quarters of single cows (n = 44).

Clinical severity score data were obtained for 75% (33/44) of the *Prototheca*-positive cows. Of these, mild (score 1) and moderate (score 2) clinical severity scores were observed in 78.8% (26/33) and 21.2% (7/33) of cows, respectively, showing abnormalities exclusively in milk and mammary glands. None of the animals showed a severe infection (score 3) or systemic signs.

Among the 44 isolates initially diagnosed as *Prototheca* (i.e., upon colony macro- and micromorphology), 84.4% (38/44) and 13.6% (5/44) were identified either as *P. bovis* or *Prototheca* sp. using MALDI-TOF MS, respectively. In contrast, PCR sequencing of the *cytb* gene allowed all 44 *Prototheca* isolates to be identified as *P. bovis*.

The in vitro algicide activity of QA was observed at low concentrations against all 44 *Prototheca* mastitis isolates and three type strains, at three concentrations: 35 ppm, 17 ppm, and 8 ppm (Table 2; Figure 2 and Figure 3).

Mammary infections caused by *Prototheca* species pose a contemporary and serious challenge to the control of bovine mastitis [37,38], particularly among modern dairy herds with well-established control measures for traditional contagious pathogens [14]. These difficulties are due to a wide distribution and adaptation of the algae to a variety of biomes [16], including milking environments and surrounding areas that favor intramammary infections [39], in addition to a lack of effective protocols for intramammary or systemic treatment of algal infections [37,40], a fact that may be considered the primary motivation of the current study.

A substantial increase in cases and outbreaks of *Prototheca*-related bovine mammary infections has recently been described in different countries, including Canada [13], Italy [11,41], Poland [1,9], Korea [12], China [10] and Ecuador [8]. In Brazil, mastitis was first described in 1992 in cows with chronic clinical mastitis in the interior of the state of São Paulo, southeastern Brazil [42]. Subsequently, the algae has been described in several states of the country with expressions in milk production, mainly in clinical, chronic cases, in rural properties with excess humidity and organic matter in the milking environment and deficiency in the practice of milking [6,7,23,24]. Currently, it is estimated that breast infections by *Prototheca* in the country occur in between 5 and 10% of clinical cases, and have been considered one of the main agents of cases of environmental mastitis [5]. In the present study, of the 4275 milk samples from clinical mastitic cows, the algae were isolated among 1% of the animals, with the within-farm prevalence not exceeding 5%. The low prevalence of clinical mammary protothecosis among dairy herds in this study can be credited to sound knowledge and strict adherence to hygienic milking practices preventing the spread of environmental and contagious pathogens [16].

Clinical severity scoring of cases has been investigated in bovine mammary infections [17], in addition to a set of virulence factors from some pathogens, especially of Enterobacteriaceae origin [19], to gain a better understanding of the pathogenic mechanisms involved in these infections, aiming to adopt proper preventive and control measures and to develop therapeutic interventions targeting the rational use of antimicrobials in dairy herds [43]. Among cows for which clinical severity scoring was available, 78.8% (26/33) and 21.2% (7/33) had mild and moderate infections, respectively. Conversely, no animal showed severe clinical signs, reinforcing the view that bovine mammary protothecosis induces abnormalities preponderantly in milk and mammary gland tissue [16], and rarely manifests with systemic signs [44]. To our knowledge, our study is the first to apply the clinical severity scoring of bovine mammary infections by *Prototheca*.

Routine diagnosis of protothecal mammary infections in cattle has been performed using milk cultures (farms and laboratories), identification of the algae based on macro- and micromorphology features of the cells, and biochemical testing (e.g., carbohydrate assimilation profile) [1,45]. Recently, a new molecular marker targeting the mitochondrial *cytb* gene has been proposed for the reliable identification of *Prototheca* species [1]. In the present study, all 44 *Prototheca* isolates collected from clinical mammary infections were identified as *P. bovis* using PCR-sequencing of the *cytb* marker, confirming *P. bovis* as the major etiological agent of protothecal bovine mastitis [6,7,8,9,10].

The identification of *Prototheca* has also been facilitated with the introduction of MALDI-TOF MS technology [7]. In the current study, MALDI-TOF MS allowed a prompt diagnosis of *P. bovis* in 86% (39/44) of the algae isolates, while the remaining five isolates were identified as *Prototheca* sp. This inability of species discrimination might refer to some technical inaccuracies in the sample. Nonetheless, the implementation of molecular methods into the diagnosis pathway of *Prototheca* mastitis has enabled a valuable decision-making process and delivery of proper control measures in affected herds [37,40].

The lack of effective intramammary or systemic therapy for *Prototheca*-induced infections has prompted several in vitro studies involving a great variety of different compounds with potential algaecide activity [6,15,20,21,22,23,24,25,26,28,29,30]. However, only a few of the compounds approached have been assessed in vivo and provided only a temporary resolution of clinical signs [37,40]. The refractoriness of *Prototheca* species to conventional treatment under in vivo conditions can be attributed to the development of pyogranulomatous reactions [46], high genetic diversity [40], and mechanisms of virulence and evasion of the immune response [47], such as biofilm production [23,24] and the induction of cellular apoptosis in mammary cells [48,49].

The in vitro algicidal effect of QA, a sanitizer with a well-known microbicidal potential, was investigated here on 44 *Prototheca* mastitic isolates. The in vitro algicidal activity of QA was observed against all isolates, at low concentrations (i.e., within the range of 8 ppm to 35 ppm, including three type strains of *P. bovis*, *P. blaschkeae*, and *P. ciferrii*). The biocidal effect of QA has been attributed to the direct action of the drug on the plasma membrane, leading to protein denaturation, membrane rupture, and consequent death of the pathogen [33,34]. QAC has long been used a sanitizer for human and animal safety (i.e., hospital, agricultural, and industrial environments) [36], due to its good stability, low toxicity, lack of odor, low corrosiveness, and biodegradability [35].

A similar study with 106 isolates of *P. bovis* (formerly *P. zopfii* 2) also investigated the algicidal effect of QA (3%) at three different time and temperature ratios (62 °C/30 s; 72 °C/15 s; and 80 °C/10 s), and observed a high algicidal effectiveness of quaternary ammonium (88–90%) at a dilution of 1:100, indicating that these compounds could be used as sanitizers for the control of protothecal mastitis [50]. Here, the in vitro algaecide activity of QA used as a sanitizer for swimming pools was observed at low concentrations against *P. bovis*, isolated from clinical mastitic cows. Our results suggest that QA could be used at low concentrations as a sanitizer for the milking environment and surrounding areas of dairy farms, particularly those with persisting protothecal mastitis. In addition, it would be worth testing the algaecide effect of QA as pre- and post-dipping antiseptic solutions to milking cows since this compound exhibits low toxicity and causes low irritation to the mucous membranes [36].

*Prototheca* algae have zoonotic potential [51]. This applies particularly to *P. bovis* [1] and *P. wickerhamii* [5] species in terms of livestock and companion animals. Milk from infected animals should not be consumed by humans or offered to calves, due to the possibility of developing enteric infections [25,37], in addition to severe systemic disorders, mostly in immunosuppressed patients [51]. In general, since *Prototheca* algae are transmitted from cows to humans by the consumption of milk and its derivatives, they are relevant to human health [38] and require attention to adopt measures aimed at the prevention and control of protothecal mammary infections in herds.

Convenience sampling of *Prototheca* isolates and the lack of sampling of milking devices and immediate cow surroundings, which would allow for the environmental impact on the occurrence of clinical protothecal mastitis to be assessed, should be considered the main limitations of the study.

Overall, the predominance of *P. bovis* among cows with clinical protothecosis, supports the leading role of this species in the etiology of algal bovine mastitis. QA used in swimming pools was revealed to have an in vitro algaecide effect in the *Prototheca*, at low concentrations. This study contributes to the molecular epidemiology of protothecal mastitis and clinical severity scoring of cases, and explores the in vitro activity of QAC sanitizers against *Prototheca*.

## 3. Material and Methods

### 3.1. Animals and Farms

A convenience sampling of clinical mastitis from ten large dairy farms in Brazil was carried out, of which six were located in the south of the State of Minas Gerais (farms A, B, C, D, E, and F) and four in the southwest region of the State of Sao Paulo (farms G, H, I and J). 

Farms and cows enrolled in the study were eligible if they met the following inclusion criteria: (1) mastitis control programs with data recording available in management software, (2) bulk tank milk somatic cell count <400,000 cells/mL, (3) Holstein or crossbreed Holstein cows, (4) >20 L milk/cow/day, (5) a minimum of 200 lactating cows, (6) mechanical milking system, and (7) history of clinical mastitis.

### 3.2. Diagnosis of Clinical Mastitis and Severity Scores

Clinical mastitis cases were classified according to severity scoring as mild (score 1), moderate (score 2), and severe (score 3). Mild cases presented macroscopic abnormalities in milk appearance (e.g., flakes, pus, blood). Moderate cases were recognized as abnormal aspects of milk and udder signs of inflammation (i.e., redness, swelling, pain, abscesses, nodules), while additional signs of fever, inappetence, tachycardia/tachypnea, or decubitus were identified as severe cases [17]. Before the onset of the study, milkers were trained to distinguish the three severity scores of clinical cases [19]. 

Clinical milk abnormalities were observed using the strip cup test or after depositing the first jets of milk on a black rubber floor.

### 3.3. Milk Sampling

A total of 4275 milk samples from different cows with clinical mastitis from ten farms studied between 2017 and 2022 were included in this study. After milkers carried out routine pre-milking procedures (i.e., stripping, pre-dipping, and drying of teats using paper towels), antisepsis of the teat end region was performed using 70% alcohol solution. Then, approximately 15 mL of milk was aseptically collected in individual sterile plastic vials and immediately refrigerated (4–8 °C). On each farm, milk samples were kept frozen (−20 °C) for further transport to the laboratory for microbiological culture. 

Approximately 10 µL of each milk sample was plated onto defibrinated sheep blood agar (5%) and MacConkey agar (Oxoid™, Basingstoke, UK). The plates were incubated at 37 °C under aerobic conditions for 72 h [52]. Colonies suggestive of *Prototheca* (i.e., irregular to mucoid, white-to-gray, nonhemolytic yeast-like aspect, and 1–2 mm in diameter) [1,5] were subjected to Gram and Diff-quick staining for micromorphology examination. Compatible isolates (i.e., presence of spherical to oblong or wedge-shaped radially arranged sporangiospores) [5] were stored in brain and heart infusion broth (BHI—Oxoid™, Basingstoke, UK) with glycerol 85% (Merck™, Darmstadt, Germany) at −80 °C, until required for further molecular diagnostic purposes. 

Intramammary infection was defined as at least 3 CFU of *Prototheca*. Milk samples that yielded more than 3 different colony types were considered contaminated and were discarded [52].

### 3.4. Matrix-Assisted Laser Desorption Ionization-Time of Flight Mass Spectrometry (MALDI-TOF MS)

All compatible *Prototheca* isolates were identified by MALDI-TOF MS [53], with some modifications in the extraction of proteins [7]. Briefly, 20–40 μL of acetonitrile (100% P.A.) was added to each sample in the same proportion of formic acid 70% (1:1), and centrifuged. Then, 1 μL of the solution for each sample was added to the MSP steel target plate (MSP 96 polished-steel target; Bruker Daltonik™, Bremen, Germany) and allowed to dry for 20 min at room temperature. Next, the samples were recovered with 1 μL of matrix solution (2-cyano-4-hydroxycinnamic acid diluted with 50% acetonitrile and 2.5% trifluoroacetic acid). Finally, the steel target plates were placed in the equipment (Microflex LT/SH MALDI-TOF MS; Bruker and Daltonik™, Bremen, Germany).

The spectra were obtained between the 2000 and 20,000 m/z mass range using FlexControl 3.3 software. The following parameters were configured on the device: ion source 1 set at 20.0 kV; ion source 2 set at 18.2 kV; and lens set at 6.0 kV. For spectral generation, 240 laser shots of each target sample (isolate) were captured. The identification of *Prototheca* isolates was performed by MALDI-TOF MS Biotyper 4.1.70 software, supplemented with a local library, which had the insertion of *P. bovis* and *P. blaschkeae* species. 

Scores between ≥1.7 and <2.0 and ≥2.0 were considered reliable for genus and species identification, respectively [7].

### 3.5. Polymerase Chain Reaction and Sequencing of the cytb Gene 

For *Prototheca* species identification, partial *cytb* gene sequencing was performed [30]. Briefly, a loopful of *Prototheca* cells from a single colony grown on SDA agar was used for the DNA extraction procedure. Mechanical cell disruption of DNA was carried out using a purification kit (GeneMATRIX™ Environmental DNA and Amp; RNA purification™, EURx, Gdańsk, Poland) and by vigorous shaking with glass beads in a detergent-rich solution, combined with lysozyme and proteinase K action. All steps, including additional treatment with lyticase (100 g/mL) (Sigma™, Saint Louis, MO, USA) and β-mercaptoethanol (1 L/mL) (Sigma™, Saint Louis, MO, USA) were carried out according to the manufacturer’s instructions. Finally, the purified DNA, dissolved in TE buffer (10 mM Tris-HCl, 1 mM EDTA, pH 8.0), was quantified using a NanoDrop ND-1000 Spectrophotometer (Thermo Fisher Scientific™, Waltham, MA, USA) and stored at 20 °C until required for further procedures. 

For PCR amplification and sequencing, the *cytb* partial gene was amplified in 30-μL reaction mixtures containing 18 μL of Color Taq PCR MasterMix (EURx™, Gdańsk, Poland), 1 μL (ca. 10 ng) of template DNA, and 1 μL of each primer *cytb*-F1 and *cytb*-R1 (0.2 μM each). PCR was performed based on the following cycle conditions: 3 min of initial denaturation at 95 °C, followed by 35 cycles of 30 s at 95 °C, 30 s at 50 °C, and 30 s at 72 °C, with a final extension period of 5 min at 72 °C. The amplified products were visualized using agarose gel electrophoresis (1%, wt/vol) and stained with ethidium bromide. The amplicons were purified using the Short DNA Clean-up DNA purification kit (EURx™, Gdańsk, Poland) and directly sequenced with the same primers used for PCR amplification.

The resulting sequences were assembled in the Clone Professional Manager Suite 8 program. The results were then compared with Blast (https://blast.ncbi.nlm.nih.gov/Blast.cgi?PROGRAM=blastn&PAGE_TYPE=BlastSearch&LINK_LOC=blasthome accessed on: 21 October 2022) *Prototheca* ID databases (https://prototheca-id.org accessed on: 21 October 2022) [54]. 

### 3.6. In Vitro Algaecide Activity of QA 

In vitro algaecide activity of QA against *Prototheca* isolates was performed as described previously using other sanitizers [6,26].

Two to three *Prototheca* colonies of each isolate were inoculated in tubes containing 3 mL of BHI (Oxoid™, Basingstoke, UK) and incubated in aerobic conditions at 37 °C for 48 h. The turbidity of the tubes was adjusted by optical density equivalent to the 0.5 standards of the McFarland scale using a sterile 0.9% saline solution [6,26]. For the evaluation of the algaecide activity, a commercial product for swimming pools was used (Oxi Algacida^®^, Sandet Química Ltd.a., Sao José do Rio Preto, SP, Brazil) containing exclusively QA as the active compound (alkyl amido propyl dimethyl benzyl ammonium chloride) in liquid form at a 15% concentration.

For each *Prototheca* isolate, a series of 15 dilutions in separate sterile tubes was prepared, with a starting QA solution (1 mL) at a concentration of 15% or 150,000 parts per million (ppm) deposited. In the 14 subsequent tubes, 0.5 mL of sterile Milli-Q water as a diluent was distributed. Then, 0.5 mL of QA was transferred from the first to the second tube, with subsequent homogenization. This same process was carried out successively in the other 14 tubes, discarding 0.5 mL after the homogenization of tube 15. Then, 0.5 mL of the inoculum containing *Prototheca* cells was added to each tube, at the following concentrations: 75,000, 37,000, 18,000, 9000, 4500, 2250, 1120, 560, 280, 140, 70, 35, 17, 8, and 4 ppm. The tubes were then incubated overnight under aerobic conditions, at 37 °C. After that time, an aliquot of 10 µL of each tube was plated on Sabouraud agar medium and kept under aerobic conditions at 37 °C for 7 days, with readings taken every 24 h, aiming to evaluate the minimum algicidal concentration of the QA. The testing was carried out in duplicate for all isolates.

As a control, suspensions were prepared in sterilized tubes containing 0.5 mL of sterilized Milli-Q water with 0.5 mL of the inoculum, adjusted to the same turbidity (0.5 McFarland standard). The following *Prototheca*-type strains were used as references: *P. bovis* (SAG 2021), *P. ciferrii* (SAG 2063), and *P. blaschkeae* (SAG 2064).

Kaplan–Meier survival analysis was conducted to determine the antimicrobial effect of QA against *Prototheca* isolates. For this, the antimicrobial concentration was used as a time variable, and inhibition of bacterial growth was used as an event.

## 4. Conclusions

*P. bovis* was the species identified by MALDI-TOF MS and PCR in the sampled animals, reinforcing the predominance of this species of algae as the primary agent of clinical mastitis in cows on large dairy farms in Brazil. The absence of severe cases of mastitis was observed, reinforcing those infections by protothecae, particularly *P. bovis*, are mainly restricted to the mammary gland, without dissemination or systemic manifestations by the animals. Quaternary ammonium showed an algicidal effect at low concentrations on *Prototheca* isolates and can be used as an alternative sanitizer for the milking environment or in pre- and post-dipping solutions in the control/prophylaxis of mammary protothecosis.

## Figures and Tables

**Figure 1 animals-13-03286-f001:**
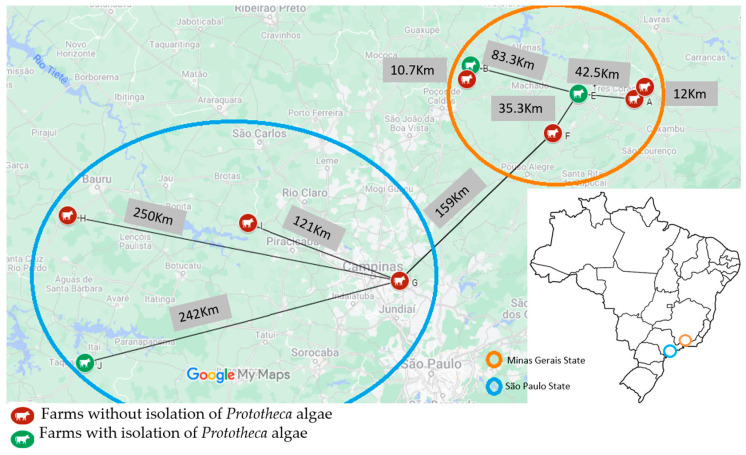
Geographical location and distance between ten dairy farms sampled in Sao Paulo and Minas Gerais states, Brazil, and isolation of *Prototheca* algae among cows with clinical mastitis mammary infections (2017–2022).

**Figure 2 animals-13-03286-f002:**
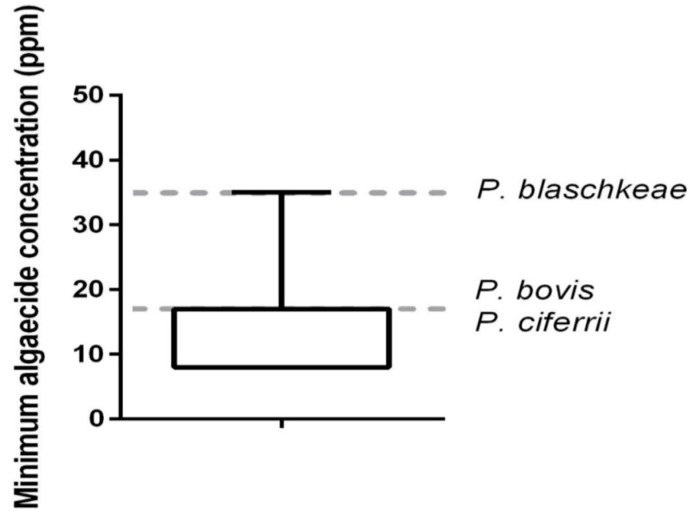
In vitro minimum algaecide concentration of quaternary ammonium on 44 *Prototheca bovis* isolated from clinical bovine mastitis and reference strains of *Prototheca* (*P. bovis*, *P. blaschkeae*, and *P. ciferrii*). Reference strains provided by Dr. Tomasz Jagielski, University of Warsaw, Poland [1].

**Figure 3 animals-13-03286-f003:**
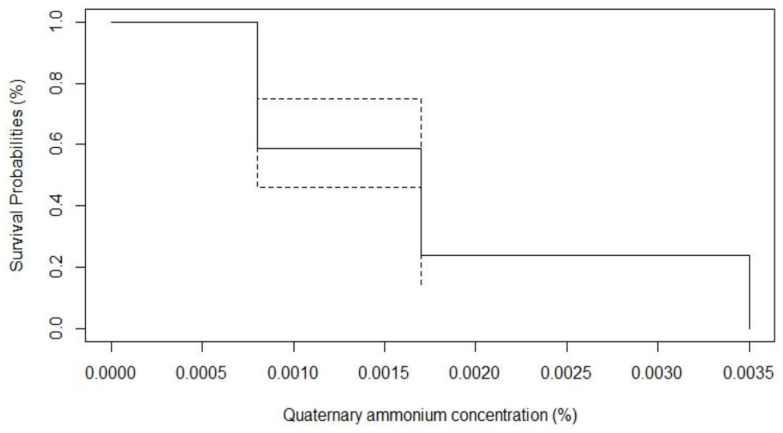
Survival probabilities of 44 *P. bovis* isolated from bovine clinical mastitis in face of in vitro algaecide effect of quaternary ammonium.

**Table 1 animals-13-03286-t001:** Frequency of *Prototheca bovis* identification based on PCR-sequencing of the *cytb* gene marker in bovine clinical mastitis among farms in Sao Paulo and Minas Gerais states, Brazil (2017–2022).

Farms	States	PCR-Sequencing of *cytb* Gene
		% (N)
B	Minas Gerais	3.7 (8/212)
E	Minas Gerais	4.6 (33/721)
J	São Paulo	4.5 (3/66)

N = number of *P. bovis*/total number of milk samples cultured; % = percentage.

**Table 2 animals-13-03286-t002:** In vitro algaecide effect of quaternary ammonium in *Prototheca* isolated from cows with clinical mastitis (2017–2022).

		Algaecide Effect		
*Prototheca* species		35 ppm (%)	17 ppm (%)	8 ppm (%)
Isolates	*P. bovis*	9/44 (20.4)	16/44 (36.3)	19/44 (43.2)
References *	*P. bovis* *	--	1/1	--
	*P. blaschkeae* *	1/1	--	--
	*P. Ciferrii* *	--	1/1	--

N = number of isolates with algaecide effect/total number of isolates; % = percentage; ppm = parts per million; * Reference strains provided by Dr. Tomasz Jagielski, University of Warsaw, Poland [1].

## Data Availability

The data presented in this study are available on request from the corresponding author.

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
