# Peer review of "In Vitro Activity of Quaternary Ammonium in *Prototheca* Isolated from Clinical Bovine Mastitis Identified by Mass Spectrometry and PCR Sequencing of the *cytb* Gene Marker"

_animals, 2023, doi:10.3390/ani13203286_

Round 1
Reviewer 1 Report
Dear authors,
The topic of the present study it is very important to improvement the Herd Health of all dairy farms around the world.
Please, I suggest that you highlight even more the conclusions in which the use of QA as a sanitizer is recommended and the health risk implied by clinical treatments of mastitis or any other infection caused by P. bovis.
There are some comments and observations in the text of attached file. Please check it.

Author Response
Dear reviewer, thank you for the suggestions made in the manuscript. We accepted and highlighted the changes in blue, including the "conclusions" topic.
Please see the attachment.

Reviewer 2 Report
The paper reports the results of a large study on the activity of quaternary ammoniun against sensitivity of several Prototheca bovis cultured from mastitic milk. The manuscript is of interest and reports interesting data on the suitability of PCR typing in comparison with MALDI-TOF, too.
Figure 2 - The Authors should specify how MIC values were achieved. Please, check the caption, also.
Table 2 should be referred to MIC values and percentages should be written into brackets. For references strains, please, omit the percentage values, being only a specimen per species)
Please avoid to repeat the term algae after "Prototheca"
The Authors should revise the English form of their manuscript
Author Response
Dear reviewer, I don't understand how you want to specify the achieved MIC value in the figure. I also ask that you specify what you want us to change in the caption.
At the end of the references, follow the annex of the certificate of revision of the English writing.
Please see the attachment.

Reviewer 3 Report
Major points
The article by Filho et al. describes the results of an epidemiological study on bovine protothecosis in Brazil. The authors analysed a large number of samples, and identified the cultured Prototheca strains according to standard methods – MALDI-TOF MS and CYTB-gene PCR-sequencing. Moreover, the authors assessed the susceptibility of the Prototheca strains to quaternary ammonium compounds. Overall, the study is well designed and the results are well presented and discussed. However, I would ask the authors to address the following issues:
- The title refers almost exclusively to the susceptibility of Prototheca to QA. In my opinion the prevalence of bovine protothecosis should be equally reflected in the title.
- The authors claim that their work is the only one to investigate the susceptibility of Prototheca to QAC. This is clearly not true, as the authors cite another work, by Lassa et al. (line 222). These previous findings should be more emphasized in the article. Moreover, in the Discussion, much more attention should be given to the results of the study by Lassa et al. For instance, the concentrations of QA from that work and the present one were not compared at all. Please, develop this paragraph and provide more details, so that the reader can easily compare your results with those from the previous study.
- The article also lacks some discussion on the epidemiology of bovine protothecosis in Brazil over the years. The results from this study should be compared with those from previous studies of that kind (in terms of Prototheca mastitis epidemiology). I am eager to know more about the changes/trends in the epidemiology of the condition in Brazil (and perhaps neighbouring countries too).
- For a better clarity of certain parts of the text, the language of the manuscript should be verified by a native speaker.
Minor points
Line 81 Please, mention what pathogens you mean?
Figure 1 should be of much better quality.
Line 112 what „which-fam” means? It should be „within-farm” I guess.
Lines 115-116 Please, be consistent in editing the name of the cytochrome b gene (cytb vs CYTB); italicize those names.
Lines 218-222 Please, rephrase the whole paragraph; „in vitro survival of QA”; „survival” refers to organisms not to compounds; also, avoid using the name „P. zopfii”; P. bovis and P. ciferrii are the valid species names.
Line 224 or 244 No use in writing „molecularly typed”; it is obvious.
Line 237 Improve grammar of the sentence.
Line 278 delete „the”
Lines 359 or 360 Be consistent in separating values from units (e.g. 0.5mL vs 0.5 mL)
The editing of both Tables (1 & 2) should be improved
For a better clarity of certain parts of the text, the language of the manuscript should be verified by a native speaker.
Author Response
Answer #1: Thanks for the comment. However, the focus of the study was the algicidal effect of QA in cows with clinical mastitis (with the assessment of severity scores) on Prototheca isolates, whose species were identified by mass spectrometry and cytb gene sequencing. As the ten farms used in the study are highly technified and have good management and control of environmental and contagious mastitis agents, they do not reflect the real prevalence of mammary protothecosis in Brazil, which is between 5-10% of clinical cases. Thus, the authors chose not to include the prevalence obtained in the title, despite the prevalence obtained in the study being discussed based on studies in other countries (Poland, Ecuador, China, Italy, Korea).
#2: Thanks for the comment. However, the authors state that this is the first study that evaluates the clinical severity of Prototheca bovis isolates obtained from cows with mastitis. It is not mentioned in the study that this is the first study that evaluates the QA in Prototheca isolates, which were pioneered by Lassa et al., 2011 (Referenced in this article. Still, according to the advisor's suggestion, details of the study by Lassa et al. (2011) were included in the discussion. However, it is not possible to establish a direct comparison between the results, since whereas, Lassa's article used a different initial concentration (3% QA), different dilutions and thermal effect of disinfectants/antiseptics, as well as differences in algae identification (phenotypic methods).
#3: A paragraph referring to the first description of breast protothecosis in Brazil, the epidemiological profile of the cases, and the expectation of current prevalence, were included in the discussion.
#4: The revised article with the reviewers' considerations were sent for review in English - American Journal Expert, USA, by artificial intelligence mode (attached at the end of the article).
Please see the attachment.

Reviewer 4 Report
In my opinion, the authors forgot to write that milk samples were also inoculated on Sabouraud's agar.
Milk, cutaneous lesions, liquor, urine, vitreous humor, feces, rectal scrapings, tracheobronchial washing, fragmenting of organs, milking-machine surfaces, and environment material from farms have been used to isolate the algae. Prototheca species may be isolated on conventional media, such as blood agar and Sabouraud agar, in aerobic conditions. The growth of algae is optimized between 25° and 37°C. Irregular to mucoid, wet to dry, white to gray or yellowish, nonhemolytic yeast-like colonies, 1–2 mm in diameter, are isolated between 2 and 5 days, in aerobic conditions, on sheep blood agar and Sabouraud agar, depending on the species of algae (Protothecosis in Animals by Marcio Garcia Ribeiro, https://www.msdvetmanual.com/generalized-conditions/protothecosis/protothecosis-in-animals)
Author Response
Thank you for the suggestion, but in screening, the milk samples were not grown on Sabouraud agar.
Please see the attachment.

Round 2
Reviewer 2 Report
the manuscript can be now accepted
Reviewer 3 Report
Accept in present form